# Knockdown of Antisense Noncoding Mitochondrial RNA Reduces Tumorigenicity of Patient-Derived Clear Cell Renal Carcinoma Cells in an Orthotopic Xenograft Mouse Model

**DOI:** 10.3390/cancers16040830

**Published:** 2024-02-19

**Authors:** Mariela Araya, Francisca Sepúlveda, Jaime Villegas, Luis Alarcón, Luis O. Burzio, Verónica A. Burzio, Vincenzo Borgna

**Affiliations:** 1Centro Cientifico & Tecnologico de Excelencia Ciencia & Vida, Santiago 8580702, Chile; mariela.araya.ortega@gmail.com; 2Center for Regenerative Medicine, Faculty of Clinical Medicine, Clínica Alemana, Universidad del Desarrollo, Santiago 7610615, Chile; fsepulvedag@udd.cl; 3Advanced Center for Chronic Diseases (ACCDiS), Santiago 8380492, Chile; 4School of Veterinary Medicine, Faculty of Life Sciences, Universidad Andrés Bello, Santiago 8370251, Chile; jaime.villegas@unab.cl; 5Urology Service, Hospital Barros Luco-Trudeau, Santiago 8900085, Chile; luisalarconmd@gmail.com; 6Department of Biological Sciences, Faculty of Life Sciences, Universidad Andrés Bello, Santiago 8370251, Chile; lburzio@gmail.com; 7Institute of Biomedical Sciences, Faculty of Medicine, Universidad Andrés Bello, Santiago 8370134, Chile; 8School of Medicine, Faculty of Medical Sciences, Universidad de Santiago de Chile, Santiago 9170022, Chile; 9Faculty of Medicine and Science, Universidad San Sebastián, Santiago 7510602, Chile

**Keywords:** clear cell renal cell carcinoma, non-coding RNA, cancer therapy, orthotopic mouse renal model, patient-derived primary cultures

## Abstract

**Simple Summary:**

The most common form of kidney cancer is clear cell renal cell carcinoma, but its treatment is challenging due to the acquisition of resistance to existing therapies. We propose an alternative potential treatment based on targeting a non-coding RNA of mitochondrial origin, termed antisense non-coding mitochondrial RNA. Supporting this strategy is the data we present in this work, using cell cultures obtained directly from primary and metastatic patient tumors, showing that the treatment causes a stall in cell proliferation in both in vitro and in vivo mouse models, and significantly improve the survival of animals. The results from this study strongly support the use of this strategy for the development of an alternative therapeutic approach for ccRCC patients and potentially other types of renal cancer.

**Abstract:**

Clear cell renal cell carcinoma (ccRCC) is the most prevalent form of renal cancer and its treatment is hindered by a resistance to targeted therapies, immunotherapies and combinations of both. We have reported that the knockdown of the antisense noncoding mitochondrial RNAs (ASncmtRNAs) with chemically modified antisense oligonucleotides induces proliferative arrest and apoptotic death in tumor cells from many human and mouse cancer types. These studies have been mostly performed in vitro and in vivo on commercially available cancer cell lines and have shown that in mouse models tumor growth is stunted by the treatment. The present work was performed on cells derived from primary and metastatic ccRCC tumors. We established primary cultures from primary and metastatic ccRCC tumors, which were subjected to knockdown of ASncmtRNAs in vitro and in vivo in an orthotopic xenograft model in NOD/SCID mice. We found that these primary ccRCC cells are affected in the same way as tumor cell lines and in the orthotopic model tumor growth was significantly reduced by the treatment. This study on patient-derived ccRCC tumor cells represents a model closer to actual patient ccRCC tumors and shows that knockdown of ASncmtRNAs poses a potential treatment option for these patients.

## 1. Introduction

Renal cell carcinoma (RCC) is a type of cancer located in the renal parenchyma. It is the 12th most prevalent cancer with over 1.2 million patients worldwide and a mortality of 1.8 per 100,000 patients [1], representing the second highest mortality of urogenital cancers in the USA [1]. RCC is classified into subtypes [2] of which Clear Cell Renal Cell Carcinoma (ccRCC) is the most prevalent, corresponding to 70–75% of all renal carcinomas [3]. This subtype is characterized by the microscopic observation of a clear cytoplasm due to the accumulation of glycogen and lipids which dissolve during histological treatment [4]. One of the main problems of ccRCC in advanced stages is its resistance to chemo- and radiotherapy [5,6]. The development of targeted therapies [7], immunotherapies [7,8] and combined therapies [9] have improved the survival rate of these patients; however, only a small subgroup is capable of responding and the results remain poor. Currently, the use of combinations of tyrosine kinase receptor inhibitors (TKIs) and immune checkpoint inhibitors (ICIs) is recommended for RCC therapy, but resistance also develops eventually [10]. Thus, the speed of the development of new therapeutic alternatives with improved efficacy and minimal side effects is of the essence.

We described a family of mitochondrial non-coding RNAs (ncmtRNAs) derived from the 16S rRNA gene of mitochondria comprising sense (SncmtRNA) [11,12] and antisense (ASncmtRNA) [12,13] members. These transcripts are differentially expressed in human and mouse cells according to proliferative status, where SncmtRNA is expressed in all proliferating cells and ASncmtRNAs, on the other hand, are highly expressed only in normal proliferating cells, whilst tumor cells display the downregulation of these transcripts [12,13,14,15,16]. We have shown that the knockdown of ASncmtRNAs with a chemically modified antisense oligonucleotide (ASO-1537) exerts detrimental effects on tumor cells from varying tissues origins, including proliferative arrest, inhibition of metastatic potential and, ultimately, massive apoptotic death [12,14,15,16,17], while the same treatment on normal cells is essentially innocuous [12,14,15,16]. Moreover, in vivo syngeneic and xenograft murine models inoculated with several different tumor cell lines have shown a reduction in tumor growth and metastasis upon ASncmtRNA knockdown [12,14,16,17]. Based on our results, the detrimental effects of ASncmtRNA knockdown on tumor cells seems to constitute a generalized phenomenon, regardless of tissue origin.

A major part of our previous work has been focused on tumor cell lines, with similar outcomes in each case. The problem with cell lines is that through successive passages after the establishment of these lines the genomic instability, characteristic of malignant cells, induces the accumulation of countless genetic and epigenetic alterations, leading to phenotypes that hardly reflect the original tumor cells and could thus result in biased studies. However, based on our previous work, we hypothesized that ASncmtRNA knockdown would be equally efficient in primary tumor cells from ccRCC patients and we consequently aimed to establish cellular ccRCC models which could mimic more closely the genetic and physiological makeup of patient tumors. In this work, we present the study of ASO-1537 treatment on primary cultures of patient-derived ccRCC cells, both in vitro and in vivo. We show that these cells are affected in the same manner as tumor cell lines, including in an orthotopic xenograft pre-clinical model of patient-derived ccRCC cells. We also show that ASO-1537 increases survival in a subcutaneous “PDX-like” mouse model in a manner comparable to the targeted therapy drug Sunitinib.

## 2. Materials and Methods

### 2.1. Primary Cultures

The study was conducted according to the guidelines of the Declaration of Helsinki and approved by the Ethics Committee of the Barros Luco-Trudeau Hospital, South Metropolitan Health Service (Servicio de Salud Metropolitano Sur) from Santiago, Chile. Four primary and two metastatic ccRCC tumor samples as well as two samples from the opposite kidney pole from ccRCC tumors (normal tissue) were obtained from stage IV metastatic ccRCC patients, at the Barros Luco-Trudeau Hospital in Santiago, Chile, under informed consent. Tissues were minced and washed in PBS, followed by enzymatic digestion (1 mg/mL collagenase I, 1 mg/mL collagenase II, 2 mg/mL collagenase IV, 1 mg/mL dispase, 20 μg/mL hyaluronidase, 2000 U/mL DNase I and 2 mM CaCl_2_ in DMEM (Thermo Fisher Scientific, Waltham, MA, USA) at 37 °C for 45 min under constant mixing. Cell suspensions were centrifuged at 1200 rpm for 5 min and the pellet was washed in PBS, followed by another centrifugation step at 200× *g* for 5 min. Suspensions were then seeded into collagen I-coated T75 culture flasks (Thermo Fisher Scientific) in complete medium: RPMI (Thermo Fisher Scientific) supplemented with 10% fetal bovine serum (Thermo Fisher Scientific), 50 μg/mL hydrocortisone and 1X Insuline-Selenium-Transferrin supplement (Thermo Fisher Scientific) and cultured at 37 °C under a 5% CO_2_ atmosphere. 

### 2.2. Transfection

Antisense oligonucleotides used in this study contained 100% phosphorothioate internucleosidic bonds (LGC Biosearch Technology Inc., Novato, CA, USA). Cells were seeded into 12-well plates (Thermo Fisher Scientific) at 40,000 cells/well and transfected on the next day with 200 nM ASO-1537 (5′ CACCCACCCAAGAACAGG) or control ASO (ASO-C: 5′ ATATACGTACACCCTCTAACCTA) and 2 μL/well LipofectAmine2000 (Thermo Fisher Scientific) following manufacturer’s recommendations, or left untreated. Cells were then cultured for 24, 48, 72 or 96 h. Cell viability was assessed through the Trypan blue exclusion method.

### 2.3. Determination of Apoptosis 

After in vitro treatments, apoptosis was detected using the Alexa Fluor 488 Annexin V/Dead Cell Apoptosis kit (Thermo Fisher Scientific) following manufacturer’s instructions. Dead cells were stained with propidium iodide (Sigma-Aldrich, St. Louis, MO, USA) and the % of apoptotic cells was measured using flow cytometry on a BD FACS Canto-II instrument (Centro Científico y Tecnológico de Excelencia Ciencia & Vida, Santiago, Chile). Apoptosis was also measured using the Dead-End Fluorometric TUNEL System (Promega, Madison, MI, USA) and the results were analyzed under an Olympus BX53 epifluorescence microscope (Centro Científico y Tecnológico de Excelencia Ciencia & Vida).

### 2.4. Western Blot 

For the determination of changes in protein levels, treated cells were lysed in RIPA buffer supplemented with protease inhibitors (Promega), followed by protein quantification with Bradford reagent (Bio-Rad, Hercules, CA, USA). Total proteins from each sample were separated on a 12% SDS-PAGE gel and transferred to PVDF membranes (Bio-Rad) with a Transblot system (Bio-Rad). Proteins were detected with rabbit primary antibodies directed to survivin (Abcam, Cambridge, UK), PARP1 (Cell Signaling Technology, Danvers, MA, USA), N-cadherin (Thermo Fisher Scientific), cyclin B1 (Abcam), cyclin D1 (BD Pharmigen, Franklin Lakes, NJ, USA) or β-actin (Abcam) and secondary peroxidase-labeled anti-rabbit IgG. Blots were revealed with the EZ-ECL chemoluminiscent substrate (Beit-Haemek, Israel) and detected in a C-Digit Blot Scanner (LI-COR Biosciences, Lincoln, NE, USA). The quantification of protein bands was performed with the imageJ software (version 1.54f; National Institutes of Health, Bethesda, MD, USA).

### 2.5. Spheroid Formation Assay

MT1 cells treated as in Section 2.2 were collected via trypsinization, counted and 15,000 cells per well were seeded into 6-well plates coated with 2% agarose. Spheroid formation was assessed at 7 days post-seeding.

### 2.6. Orthotopic Xenograft Model 

Animal studies were conducted according to the Declaration of Helsinki, following the principle of “3Rs”. Protocols were performed following the guidelines of the National Agency for Research and Development (ANID), Chile, and approved by the Research and Ethics Committee of the “Centro Científico y Tecnológico de Excelencia Ciencia & Vida”, Santiago, Chile. NOD/SCID mice were purchased from the Jackson Laboratory (Bar Harbor, ME) and maintained at the animal facility of the “Centro Científico y Tecnológico de Excelencia Ciencia & Vida” under a 12/12 h light/dark regimen with sterile water and food ad libitum. For each experiment, ten 6–8-week-old mice (20–25 g) were inoculated with 2 × 10^6^ PT2 or MT1 cells in the subcapsular area of the left kidney. At 8–10 weeks post-cell inoculation (pci), treatments were started by intraperitoneal (ip) injections of 100 μg ASO-1537 or 100 μL saline per mouse, spaced 48 h apart. After 28 doses (17 weeks pci for PT2 and 19 weeks pci for MT1), mice were euthanized and kidneys were collected, weighed and fixed in 4% formalin for obtention of microscopy sections at the Anatomopathology Unit of the Luis Tisné Hospital (Santiago, Chile). Sections were stained in Hematoxilin/Eosin (H&E) and photographed under an Olympus BX53 microscope.

### 2.7. Subcutaneous Xenograft Model of MT1 Cells

MT1 cells were suspended at 2 × 10^6^ cells per 100 μL in a 1:1 saline–matrigel mix and injected subcutaneously on the back of an 8-week-old NOD/SCID mouse. Tumor growth was monitored with a caliper and tumor volume was estimated with the formula: length × width^2^ × 0.5236. The mouse was euthanized when the tumor reached 1000 mm^3^, which was then extracted and diced into ~30 mg-fragments, which were implanted subcutaneously into the left flank of 8-week-old NOD-SCID mice. Tumors were then overlain with 30 μL matrigel and wounds were sutured. The animals were injected i.p. with 100 μL saline or 100 μg ASO-1537 per mouse every 48 h and Sunitinib was administered orally at a dose of 30 μg/Kg daily, except for weekends. Tumor growth was monitored as above, and the mice were euthanized when tumors reached 1000 mm^3^; survival was determined using the Kaplan–Meier method.

### 2.8. Graphs and Statistical Analysis 

Data were plotted and analyzed using the Graphpad Prism 6 software. In vitro experiments were performed in triplicate with 3 independent biological replicates and analyzed using Student’s *t*-test. For in vivo experiments, comparisons between groups were performed with a non-parametric one-sided Mann–Whitney test. Survival analysis was performed using the Log-Rank (Mantel–Cox) test. Statistical significance was set at the nominal level of *p* < 0.05 or less.

## 3. Results

### 3.1. Primary Cultures of Human ccRCC

In order to explore the effects of ASncmtRNA knockdown on patient-derived ccRCC cells, we first established primary cultures from four primary tumors (named as PT1-4) and two metastatic tumors (MT1 and 2) from stage IV ccRCC patients, collected at the Urology Service of the Barros Luco-Trudeau Hospital in Santiago, Chile. All six cultures displayed the characteristic clear cytoplasm typical of ccRCC cells (Figure 1).

### 3.2. Knockdown of ASncmtRNA Induces Apoptotic Death of Primary ccRCC Cultures

We studied the effect of transfection of ASO-1537 (vs. ASO-C and non-treated cells) on the six primary ccRCC cultures (Figure 2a). We found that the treatment induces between 70 and 80% cell death at 48 h in all cultures, compared to 8–20% for control cells (Figure 2a), which is in line with our previous results on a wide array of tumor cell lines [12,14,15,16,17]. In order to confirm the selectivity of this treatment, we also established two primary cultures (NC1 and NC2) from normal kidney areas of two ccRCC patients (Appendix A), which were transfected with ASOs (or left untreated). Appendix A shows that ASO-1537 elicited a significant but modest increase in cell death, even at the highest concentration used (300 nM), reaching around 10% (NC1) and 15% (NC2), respectively. These values are very far from the 70–80% death observed for PT2 and MT1 tumor cells (Figure 2a) and thus confirm the selective nature of the knockdown treatment reported for several other cell types [12,14,15,16]. 

For further studies, we selected one primary tumor culture (PT2) and one metastatic tumor culture (MT1). The transfection of both cultures with ASO-1537 induced a high degree of apoptosis as determined through Annexin V binding assay, compared to basal levels in controls. These levels were comparable to those obtained in cells treated with the pan-kinase inhibitor staurosporin (STP) (Figure 2b). We further demonstrated apoptotic death in the MT1 culture by DNA fragmentation (TUNEL) assay (Figure 2c), reduction in survivin levels and PARP1 cleavage (Figure 2d). In addition, we compared the results on tumor cells with the first-line ccRCC treatment drug Sunitinib [18]. First, we established the IC50 of this drug for both cultures, PT2 and MT1, in order to define a range of concentrations to be used in cultured cells (Appendix A). Cells were then treated at concentrations close to or over the IC50 of each cell and compared to ASO-C or ASO-1537-transfected or non-treated cells, for 48 h. Appendix A shows that Sunitinib, at two different concentrations, displayed a lower efficiency in inducing cell death of PT2 and MT1.

### 3.3. Knockdown of ASncmtRNA Reduces Proliferative Capacity of Primary ccRCC Cultures

We reported that tumor cell lines from a wide array of human tissues undergo a proliferative block upon knockdown of ASncmtRNAs [12,14,15,16,17]. Therefore, we studied whether the patient-derived primary culture of metastatic ccRCC (MT1) would follow the same tendency. As depicted in Figure 3a, MT1 cells transfected with ASO-1537 suffered a sudden proliferation blockage, similar to STP-treated cells, while control cells continued to proliferate. This cell cycle arrest is evidenced at the molecular level through a loss in cyclins B1 and D1 (Figure 3b–d). In addition, this metastasis-derived primary culture also displays a strong reduction in N-cadherin, representing a loss in EMT potential, a key event in metastasis (Figure 3b,e). Metastatic potential is also directly related to the presence of cancer stem cells (CSCs) in the tumor population [19]. We found that ASO-1537 precluded the formation of spheroids, a hallmark of stemness in tumor cells (Appendix A).

### 3.4. Knockdown of ASncmtRNA Reduces Growth of Orthotopic Tumors Generated with ccRCC Primary Cultures

We established an in vivo orthotopic model of ccRCC primary cultures by injecting PT2 cells into the subcapsular region of the left kidney of NOD/SCID mice (Figure 4a). At 6 weeks post-cell inoculation (pci), tumors were already visible and displayed the characteristic clear cytoplasm of ccRCC (Appendix A, left panel). For the analysis of tumor growth, treatment was initiated at 8 weeks pci by ip injection every 48 h of 100 μg ASO-1537 or 100 mL saline, with a total of 28 doses at 17 weeks, where mice were euthanized and kidneys were collected, weighed and fixed. The inoculated (left) kidneys displayed larger tumors in saline-injected mice compared to ASO-1537-treated animals (Figure 4b), as evidenced by tumor weight, which was over twice as higher in the saline control group (Figure 4c).

We then performed the same analysis with the MT1 metastasis-derived primary culture (Figure 5a), which also generated tumors at 6 weeks pci (Appendix A, right panel). The same treatment regimen was applied, except for the commencement of treatment (10 weeks pci) and mice were euthanized at week 19 (Figure 5a). After necropsy, the tumors in the left kidneys of saline-treated mice were again larger than tumors from the ASO-1537 group (Figure 5b), with an average weight around 2.5 higher in the control group (Figure 5c). 

### 3.5. Comparison of ASO-1537 and Sunitinib in a Subcutaneous Xenograft ccRCC Model

In order to compare ASO-1537 with Sunitinib in vivo, we established a subcutaneous model for monitoring cell growth and thereby determining survival of tumored mice. For this model, we first generated an MT2 subcutaneous tumor in a NOD/SCID mouse, after which the tumor was extracted and diced into ~30 mg fragments which were implanted subcutaneously into other mice, followed by ASO-1537 or Sunitinib treatment, or both, using saline as the negative control group. As shown in Appendix A, both ASO-1537 and Sunitinib improved the survival of mice compared to saline, but there was no significant difference between these two treatments and the combination of both showed a small increase in survival compared to ASO-1537 alone.

## 4. Discussion

As stated above, treatment for ccRCC has proven challenging mainly due to the development of resistance to existing therapies. Therefore, new alternative treatments are urgently needed. In this work we show that, similar to our previous work in a number of cell lines from different types of cancer, primary cultures obtained from stage IV ccRCC patient tumors respond to ASncmtRNA knockdown by ceasing to proliferate and dying through apoptosis. The novelty of the present study is that primary tumor cultures mimic more accurately the genetic makeup of real tumors and should thus reflect an eventual clinical response more precisely than tumor cell lines. This is especially true for the in vivo model utilized in our study, in which we inoculated the primary cells into their tissue of origin, reflecting further similarity to a clinical scenario.

As depicted in Figure 2a, all six primary cultures derived from primary and metastatic sites show a massive degree of cell death induced by the treatment. This result is important, since metastatic tumors tend to be more refractory than primary tumors with conventional treatments, again, due to acquisition of resistance. Also, it is essential that cancer treatments kill tumor cells through apoptosis as opposed to necrosis, which is also the case in our study, as shown for the metastasis-derived MT1 cells (Figure 2b–d) and a wide variety of tumor cell lines [12,14,15,16,17]. The selectivity of the treatment was corroborated in two primary cultures of non-tumor renal epithelial cells, which were only marginally affected by the knockdown treatment (Appendix A) as observed before in several normal human and mouse cell types [12,14,15,16], which include normal kidney epithelia [14,15]. Thus, a therapy for ccRCC based on this strategy should have minor side effects, in contrast to first-line therapies such as cytokines and should therefore ensure a better quality of life for patients undergoing treatment. 

Prior to this massive cell death, as also observed before for tumor cell lines [12,14,15,16,17], there is an immediate arrest in proliferation (Figure 3a) which is also reflected in the lower growth rate of tumors in the in vivo models (Figure 4 and Figure 5). This decreased proliferation is underpinned by a strong reduction in the cell cycle factors cyclin B1 (Figure 3b,c), cyclin D1 (Figure 3b,d) and also survivin (Figure 2d), which is not only an anti-apoptotic protein but is also involved in cell cycle progression during the M-phase [20]. These results are in line with those reported previously by our group, in which we have also shown in other cell models that additional cell cycle progression factors are also downregulated through ASncmtRNA knockdown, such as CDK1, CDK4 [17], Aurora Kinase A and Topoisomerase IIα [21]. The latter two are involved in ensuring genomic integrity throughout the cell cycle [22,23] and their expression is inversely modulated in breast epithelial cells through ASncmtRNA knockdown, i.e., upregulated in normal cells and downregulated in tumor cells. We hypothesize that this inverse behavior could lie at the basis of the selectivity since it should offer genomic protection for normal cells and should, conversely, stimulate genomic instability in tumor cells in a manner which induces massive apoptotic death. In addition, the metastasis-related protein N-cadherin, an adhesion protein also involved in proliferation [24,25], is also strongly downregulated by the treatment (Figure 3b,e). N-cadherin plays a role in epithelial-mesenchymal transition (EMT), an essential first step in metastasis, suggesting a reduction in invasive potential as shown in previous reports [12,14,16]. Indeed, one of these studies was performed in the RenCa murine renal carcinoma model, in which ASncmtRNA knockdown with an ASO targeting the mouse ASncmtRNAs significantly reduced tumor growth and metastasis to the lungs in a syngeneic orthotopic model [14]. Also, as mentioned above, normal cell cultures are not damaged by the treatment but we have also observed in in vivo models an absence of damage to healthy tissue and of inflammatory response [12]. We have hypothesized that the molecular effects elicited via ASncmtRNA knockdown could be underlain by an increase in mitochondrial microRNAs (miRNAs) contained in ASncmtRNA-2, induced through the knockdown treatment [15,17,26]. These miRNAs could putatively target the key genes that are affected by the treatment, thus causing the effects on proliferation and cell death that we have observed in several tumor cell models.

Our in vivo orthotopic results (Figure 4 and Figure 5) showed that ASO-1537 effectively elicited an antitumor effect, reflected in reduced tumor weight at the endpoint of the experiments. This result implies that the ASO can be taken up by the tumors, in this case, ccRCC. Indeed, we previously reported, in a mouse orthotopic renal carcinoma model, that a fluorescently labeled ASO corresponding to the mouse counterpart of ASO-1537 is effectively incorporated into the subcapsular tumor tissue [14]. Nevertheless, the optimization of delivery of ASOs such as this to specific target tissues for cancer and other diseases remains a challenge in clinical development, with several different delivery strategies under development [27]. However, we have shown here a proof-of-concept study on the ability of ASncmtRNA knockdown to preclude the proliferation and reduce viability of primary ccRCC cells from patients.

We aimed to determine if our treatment would be comparable to existing therapeutic approaches for ccRCC. Of the plethora of targeted drugs against ccRCC [3], we chose to compare ASO-1537 with Sunitinib (SU011248), an oral small molecule TKI that exhibits potent antiangiogenic and antitumor properties and constitutes a first-line treatment for ccRCC [18,28]. In cultured PT2 and MT1 cells, Sunitinib displayed a lower efficiency than ASO-1537 at inducing cell death (Appendix A). To perform this comparison in vivo, we used a subcutaneous model to monitor tumor growth up to an ethic endpoint and to determine survival rates, which is the ultimate measure of the efficacy of a treatment. In addition, we chose to use a “PDX-like” approach. A PDX (patient-derived xenograft) is an in vivo model in which small fragments of patient tumors are directly implanted into immunocompromised mice, followed by experimental treatments [29]. In our approach, the patient tumors were cultured, injected into a mouse for tumor formation and then the fragments of that tumor where implanted. The aim of this assay was to mimic more closely an actual tumor, albeit non-orthotopic. The results shown in Appendix A evidenced an increased survival rate for both ASO-1537 and Sunitinib, with no significant difference between them. However, a combination of both elicited an apparent slight increase in survival, which suggests that there could be a small additive (but not synergistic) effect by combining these two therapies. 

But if this is not the case, what would the advantage of ASO-1537 over Sunitinib (or other small molecule drugs) be? As mentioned above, drugs such as Sunitinib have shown promising outcomes in early clinical studies [30,31], but most patients will develop resistance to Sunitinib after several months of treatment [32,33] as has also been the case for combination therapies using TKIs and ICIs [10]. To the best of our knowledge, there are no reports on development of resistance to ASO therapy, which is expected due to the mechanism of action and targets (RNAs) of this type of drug [27]. In addition, ASOs have been shown to be well-tolerated, with few off-target effects and should thus not elicit severe side effects [27]. Indeed, ASO-1537 (then known as Andes-1537) underwent a Phase Ia Clinical Trial (NCT02508441 where no significant toxicity was encountered, even up to the highest dose utilized (800 mg/Kg) [34]. Hence, the most noteworthy impact of this approach would lie in its tolerability and lack of resistance development, thereby improving the efficacy and quality of life of patients undergoing therapy.

Another instance of resistance to small molecule drugs is the presence of cancer stem cells (CSCs) within the tumor cell population, including ccRCC [10]. These cells are usually refractory to classical antitumor drugs, due to the surface expression of ABC transporters which expel small molecules [35]. To date, we have not directly studied the effects of ASncmtRNA knockdown on CSC-enriched populations, but we have observed the same expression profile of ncmtRNAs as bulk tumor cells (i.e., ASncmtRNA downregulation). An assay that is related to stemness is spheroid formation in anchorage-independent conditions. It has been shown that a primary culture from a stage IV ccRCC patient readily forms spheroids in which most cells express stemness markers [36]. As previously reported, ASncmtRNA knockdown reduces spheroid formation capacity in different tumor cell models [12,16]. Here, we show that this is also the case for MT1 cells (Appendix A) and thus ASO-1537 should hypothetically target CSCs as well as bulk tumor cells, but this hypothesis warrants further research.

In order to delve deeper into the effects of the treatment in a more realistic scenario, further preclinical work will involve screening cohorts of “real” ccRCC PDX tumors with assessment of tumor growth and metastasis concomitantly with a molecular analysis of any possible effects of ASO-1537 on healthy tissue, although ASO-1537 has already proven to be well-tolerated in human subjects [34] as described above. 

Together with our prior body of evidence in tumor cell lines, the results presented here on a cell model representing a closer similarity to a clinical scenario support the application of ASncmtRNA knockdown as a treatment strategy for ccRCC and likely other types of renal cancer. 

## 5. Conclusions

The work presented here shows that primary cultures obtained from both primary and metastatic tumors obtained from stage IV ccRCC suffer proliferative blockage and apoptotic death after the knockdown of ASncmtRNAs, as we had observed before in tumor cell lines. In addition, the treatment precludes the growth of orthotopic xenograft tumors in mice and increases survival. These results are very promising, since they reflect a closer reality to actual patient tumors as opposed to established tumor cell lines and potentiate the use of this technology for a treatment alternative against ccRCC which commonly presents resistance to other therapeutic strategies.

## Figures and Tables

**Figure 1 cancers-16-00830-f001:**
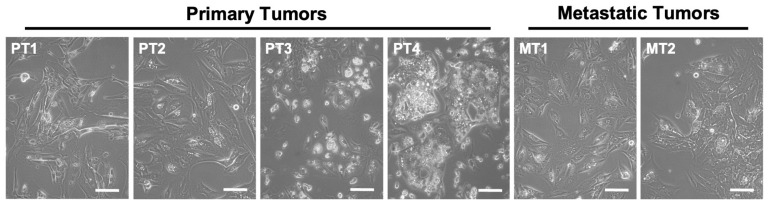
Phase contrast images of primary cultures derived from ccRCC primary tumors (PT1-4) and metastatic tumors (MT1 and 2) showing the characteristic clear cytoplasm of ccRCC cells (arrows). Bars = 100 μm.

**Figure 2 cancers-16-00830-f002:**
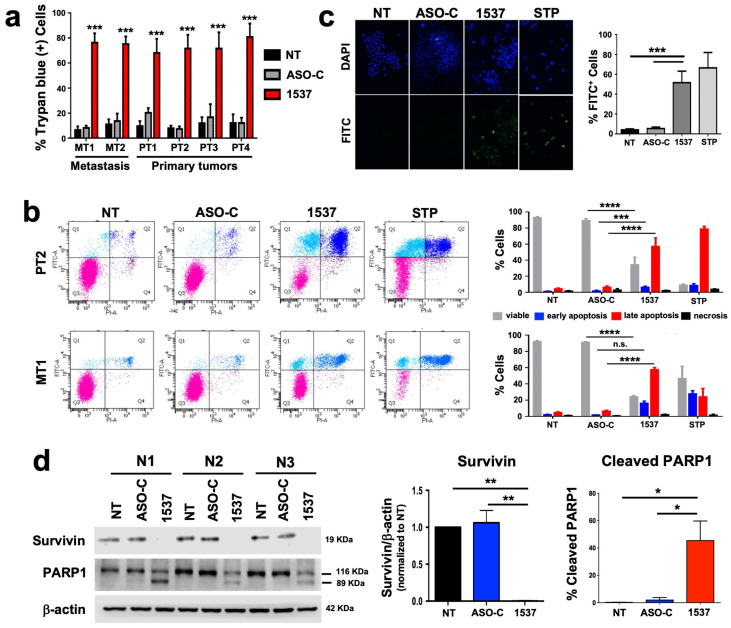
ASncmtRNA knockdown induces apoptotic death of patient-derived ccRCC cells. Cultures were transfected with ASO-1537 or ASO-C, or left untreated (NT), for 48 h. (**a**) Cell death was evaluated in all six primary cultures through Trypan blue exclusion, asterisks indicate significant difference in ASO-1537-treated cells compared to both controls (NT and ASO-C); (**b**) Apoptosis was determined through Annexin V binding in PT2 and MT1 cells. Left panel, dot plot of a representative flow cytometric analysis of cells under different conditions and using staurosporin (STP) as a positive control. Cell populations are marked with colors: pink, viable; cyan, early apoptosis; blue, late apoptosis; purple, necrosis. Right panel, graphic representation of three independent Annexin V assays performed by flow cytometry as in the left panel. (**c**) Apoptosis was also determined by TUNEL assay in MT1 cells, using staurosporin (STP) as a positive control. Left, representative fluorescent microscopy images of cells treated under different conditions showing total (DAPI) and TUNEL-positive (FITC) cells; right, graphic representation of three independent TUNEL determinations; (**d**) survivin levels and % of cleaved PARP1 were determined using Western blot. * *p* < 0.05; ** *p* < 0.01; *** *p* < 0.001; **** *p* < 0.0001; n.s., non-significant. The images of original blots for (**d**) can be found in Appendix A and densitometry readings in Appendix A.

**Figure 3 cancers-16-00830-f003:**
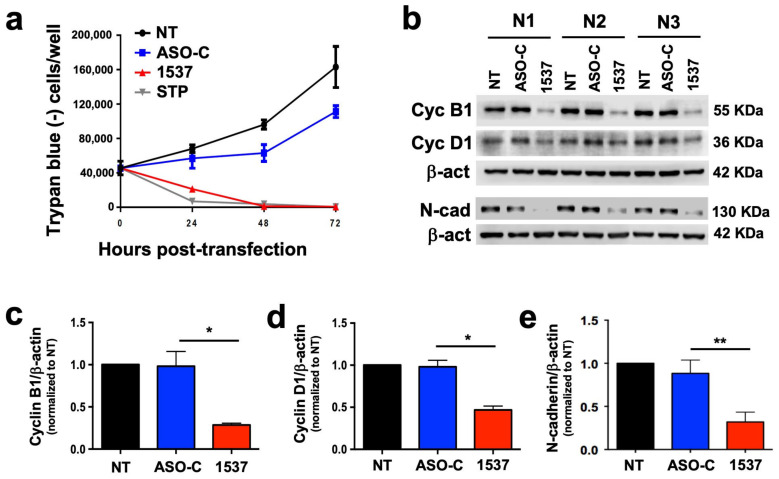
ASncmtRNA knockdown induces proliferative blockage of patient-derived ccRCC cells. MT1 cultures were transfected with ASO-1537 or ASO-C, or left untreated (NT), for 48 h. (**a**) Growth curves determined through Trypan blue exclusion; STP was used as a positive control, (**b**) cell cycle (cyclin B1 and D1) and EMT (N-cadherin) protein levels were determined using Western blot; the image shows the results of three independent experiments, which were used for quantification of cyclin B1 (**c**), cyclin D1 (**d**) and N-cadherin (**e**); * *p* < 0.05; ** *p* < 0.01. The images of original blots for (**b**) can be found in Appendix A and densitometry readings in Appendix A.

**Figure 4 cancers-16-00830-f004:**
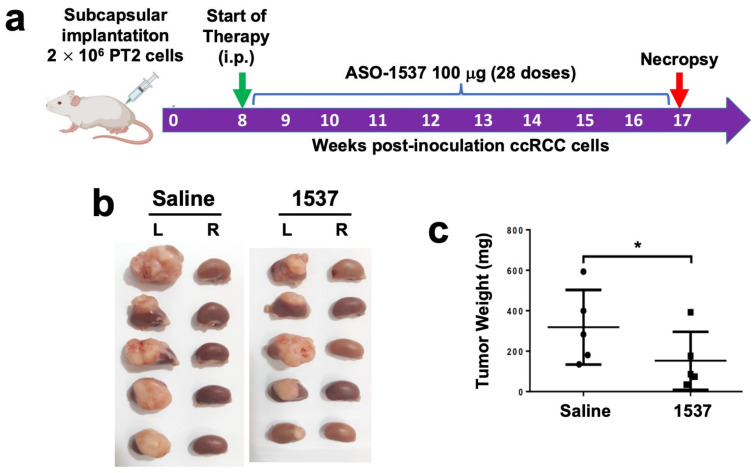
ASO-1537 reduces growth of PT2 subcapsular tumors. (**a**) Scheme depicting the protocol used for the orthotopic xenograft model. Injections were performed every 48 h and saline was used as control; (**b**) images showing the left (inoculated) and right (non-inoculated) kidneys of the five mice in each experimental group at 17 weeks post-cell inoculation; (**c**) tumor weight of the five mice in each group; * *p* < 0.05. Mouse and syringe image in (**a**) was created using Biorender (biorender.com) (accessed on 30 August 2023).

**Figure 5 cancers-16-00830-f005:**
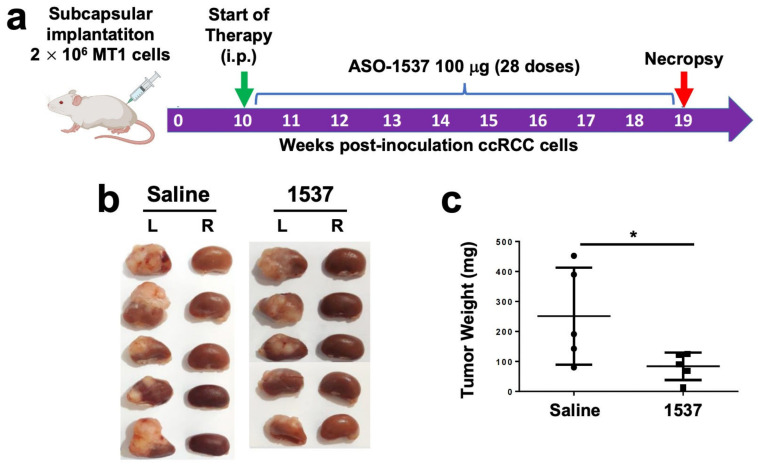
ASO-1537 reduces growth of MT1 subcapsular tumors. (**a**) Scheme depicting the protocol used for the orthotopic xenograft model. Injections were performed every 48 h and saline was used as control; (**b**) images showing the left (inoculated) and right (non-inoculated) kidneys of the five mice in each experimental group at 19 weeks post-cell inoculation; (**c**) tumor weight of the five mice in each group; * *p* < 0.05. Mouse and syringe image in (**a**) was created using Biorender (biorender.com) (accessed on 30 August 2023).

## Data Availability

The raw data supporting the conclusions of this article will be made available by the authors on request.

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
