# Peer review of "Knockdown of Antisense Noncoding Mitochondrial RNA Reduces Tumorigenicity of Patient-Derived Clear Cell Renal Carcinoma Cells in an Orthotopic Xenograft Mouse Model"

_cancers, 2024, doi:10.3390/cancers16040830_

Round 1

Reviewer 1 Report

Comments and Suggestions for Authors

The article titled "Knockdown of antisense noncoding mitochondrial RNA reduces tumorigenicity of patient-derived clear cell renal carcinoma cells in an orthotopic xenograft mouse model" primarily investigates the effects of targeting antisense non-coding mitochondrial RNA (ASncmtRNA) in clear cell renal cell carcinoma (ccRCC). The study demonstrates that ASncmtRNA knockdown induces proliferative arrest and apoptotic death in tumor cells, potentially offering a new therapeutic approach for ccRCC patients. 

Overall, the study's focus on a novel treatment approach for ccRCC, a cancer resistant to many existing therapies, is of significant interest and adds valuable knowledge to the field. The potential application of these findings could be of high interest to the scientific community, especially in the field of cancer therapy.

  • Novelty and Interest: The study appears to contribute new insights into ccRCC treatment, especially in the context of resistance to existing therapies. Its focus on patient-derived cells and in vivo models enhances its relevance.

Here are some aspects for improvement and considerations regarding its novelty and interest:

  • Writing Quality: The manuscript should be free of grammatical errors. Please check typos.
  • Introduction: Regarding the introduction of the provided article, a critical comment would be that it could benefit from a more detailed and clear presentation of the research objectives and background context. An effective introduction should offer a comprehensive overview of existing knowledge on the topic, highlighting the study's relevance and importance. It should also clearly define the research objectives and hypotheses, setting the stage for the ensuing results and discussions. This would help in orientating the reader to the specific issues addressed in the study and establish a framework for understanding its contributions to the field.
  • Discussion: In the discussion section of the provided article, a critical observation would be that it could benefit from a more comprehensive analysis and interpretation of the results. A well-structured discussion should not only interpret the findings in the context of existing literature but also critically examine the limitations and implications of the study. Additionally, suggesting potential directions for future research based on the study's outcomes could enhance the section's value. This approach would provide a more holistic understanding of the study's impact and relevance in the broader field of research. Moreover, compare this approach with recent ccRCC treatments, like those in the suggested article (PMID: 37685983).  Discuss how antisense oligonucleotides might affect cancer stem cells (CSCs) or signaling pathways, referencing the suggested article in the discussion section.
  • Potential Impact: Highlight the potential impact of the findings on ccRCC treatment and future research directions.

Author Response

The article titled "Knockdown of antisense noncoding mitochondrial RNA reduces tumorigenicity of patient-derived clear cell renal carcinoma cells in an orthotopic xenograft mouse model" primarily investigates the effects of targeting antisense non-coding mitochondrial RNA (ASncmtRNA) in clear cell renal cell carcinoma (ccRCC). The study demonstrates that ASncmtRNA knockdown induces proliferative arrest and apoptotic death in tumor cells, potentially offering a new therapeutic approach for ccRCC patients. 

Overall, the study's focus on a novel treatment approach for ccRCC, a cancer resistant to many existing therapies, is of significant interest and adds valuable knowledge to the field. The potential application of these findings could be of high interest to the scientific community, especially in the field of cancer therapy.

 Novelty and Interest: The study appears to contribute new insights into ccRCC treatment, especially in the context of resistance to existing therapies. Its focus on patient-derived cells and in vivo models enhances its relevance.

Author response: We sincerely thank the reviewer for his or her valuable insight and comments to improve the manuscript. We have addressed the concerns below:

Here are some aspects for improvement and considerations regarding its novelty and interest:

Writing Quality: The manuscript should be free of grammatical errors. Please check typos.

Author response: As pointed out by the reviewer, there were a few typos throughout the text, which have now been corrected.

Introduction: Regarding the introduction of the provided article, a critical comment would be that it could benefit from a more detailed and clear presentation of the research objectives and background context. An effective introduction should offer a comprehensive overview of existing knowledge on the topic, highlighting the study's relevance and importance. It should also clearly define the research objectives and hypotheses, setting the stage for the ensuing results and discussions. This would help in orientating the reader to the specific issues addressed in the study and establish a framework for understanding its contributions to the field.

Author response: We thank the reviewer for this suggestion. We have enriched the introduction and added the hypothesis and main aim.

Discussion: In the discussion section of the provided article, a critical observation would be that it could benefit from a more comprehensive analysis and interpretation of the results. A well-structured discussion should not only interpret the findings in the context of existing literature but also critically examine the limitations and implications of the study. Additionally, suggesting potential directions for future research based on the study's outcomes could enhance the section's value. This approach would provide a more holistic understanding of the study's impact and relevance in the broader field of research. Moreover, compare this approach with recent ccRCC treatments, like those in the suggested article (PMID: 37685983).  Discuss how antisense oligonucleotides might affect cancer stem cells (CSCs) or signaling pathways, referencing the suggested article in the discussion section.

Author response: As per this and other reviewer’s suggestions, we have expanded the results and discussion sections, including a more detailed and critical analysis of our results. As for cancer stem cells, we have not studied this area very thoroughly, we have only determined that the expression profile of CSCs is the same as bulk tumor cells (i.e. ASncmtRNA downregulation), but we have not performed studies with ASO-1537 on CSC-enriched populations. However, the observed expression profile could suggest that the treatment will affect CSC as efficiently as bulk tumor cells. We have included a spheroid formation assay in MT1 cells (new Figure S6), which is a hallmark for stemness. In this assay we observed a loss of spheroid formation in ASO-1537-treated cells. We have also included the analysis of this assay into the discussion sections, and included the reference suggested by the reviewer.

Potential Impact: Highlight the potential impact of the findings on ccRCC treatment and future research directions.

Author response: We have added comments to the discussion regarding the reviewer’s suggestions.

Reviewer 2 Report

Comments and Suggestions for Authors

Araya M et al present an interesting and clinically important study demonstrating the use of primary cultures obtained from both primary and metastatic ccRCC patient tumors as a model to develop therapies against Renal carcinoma as these cells are closer to actual patient ccRCC tumors. Further, their findings suggest that application of ASncmtRNA knockdown could be a potential treatment strategy for ccRCC. However there are certain gaps in the experiments and derived conclusions as follows-

  1. Typo in line 30
  2. What is the expression of SncmtRNA and ASncmtRNA in patient-derived ccRCC cultures and used in the study in comparison to control renal cells? An exact copy number quantification using digital PCR will be useful. 
  3. Figure 2B does not seem to have any statistical significance indicated.
  4. Figure 3A-growth curve should be repeated using another assay such as the Cel Titre Glo. The assay should also be conducted using PT2 cells.
  5. Figure 4 and 5- what is the expression of ASncmtRNA in the tumours treated with saline and those with the ASO?
  6. Figure 4- are there survival curves for PT2 tumors?

Author Response

Araya M et al present an interesting and clinically important study demonstrating the use of primary cultures obtained from both primary and metastatic ccRCC patient tumors as a model to develop therapies against Renal carcinoma as these cells are closer to actual patient ccRCC tumors. Further, their findings suggest that application of ASncmtRNA knockdown could be a potential treatment strategy for ccRCC. However there are certain gaps in the experiments and derived conclusions as follows-

Author response: We thank the reviewer for taking the time to assess our work. We have addressed the concerns raised by him or her as follows:

  1. Typo in line 30

Author response: Thank you, this and other typos we found in the MS have been corrected.

  1. What is the expression of SncmtRNA and ASncmtRNA in patient-derived ccRCC cultures and used in the study in comparison to control renal cells? An exact copy number quantification using digital PCR will be useful.

Author response: The reviewer raises a rational concern. However, we did not perform this analysis and, unfortunately, we cannot do so now since, due to an accident which caused a major crack in our liquid nitrogen tank when we moved our lab to a different location last year, we lost all of our cell cultures, including the ones used in the present study. However, we have now included a new Supplementary Figure (Fig. S1) in which we present normal renal epithelial cultures obtained from non-tumor tissue dissected from the poles opposite from the tumors of two ccRCC patients. Although we did not quantify ASncmtRNA in these samples either, we do, however, observe a minimal effect of ASncmtRNA knockdown even up to a higher concentration of ASO-1537 than that used for tumor cells. This figure shows, as in previous publications, the selective nature of the knockdown treatment with minimal effect on normal renal cells.

  1. Figure 2B does not seem to have any statistical significance indicated.

Author response: We apologize for this oversight; we have now added the statistical significance in Fig. 2b.

  1. Figure 3A-growth curve should be repeated using another assay such as the Cel Titre Glo. The assay should also be conducted using PT2 cells.

Author response: Due to the reason mentioned in the response to comment 2, we are unable to perform the experiment suggested by the reviewer. However, we consider that, although the Trypan blue exclusion assay is not as sensitive as the Cel Titre method, the curves display a sufficiently significant difference between them to conclude that ASO-1537 elicits a cell proliferation blockage when compared to the controls. We hope that this reponse is satisfactory for the reviewer.

  1. Figure 4 and 5- what is the expression of ASncmtRNA in the tumours treated with saline and those with the ASO?

Author response: Again, we regret that we are unable to perform this quantification. However, we have previously published data confirming the knockdown of ASncmtRNA after the treatment, both in vitro and in vivo (Vidaurre et al., 2014 doi: 10.1074/jbc.M114.558841; Lobos-González et al., 2016 doi: 10.18632/oncotarget.11110; Borgna et al., 2017 doi: 10.18632/oncotarget.18460; Varas-Godoy et al., 2018 doi: 10.1111/pcmr.12615; Fitzpatrick et al., 2019 doi 10.1038/s41419-019-1649-3; Borgna et al., 2020 doi: 10.7150/jca.38880). These determinations have been performed in different cell models, so the effect seems essentially generalized regardless of tissue type.

  1. Figure 4- are there survival curves for PT2 tumors?

Author response: We did not perform survival curves for the PT2 culture (obtained from a primary tumor) because we considered that, in order to reduce the number of experimental animals, we should privilege the more aggressive (metastasis-derived) MT2 cells. And again, we are not able to perform a new experiment such as this for the reasons mentioned above.  

We sincerely apologize for not being able to conform to all of the reviewer’s concerns, we hope that he or she will still consider our present work to be a valuable contribution to the field.

Reviewer 3 Report

Comments and Suggestions for Authors

The manuscript investigates the therapeutic potential of mitochondrial non-coding RNAs (ASncmtRNAs) knockdown in clear cell renal carcinoma (ccRCC). The study focuses on primary cultures derived from ccRCC patient tumors, providing a more clinically relevant model than traditional cell lines. Both primary and metastatic ccRCC, exhibit a pronounced blockage in proliferation and undergo apoptotic death upon ASncmtRNA knockdown, which were consistent with previous findings in tumor cell lines. Furthermore, ASncmtRNA knockdown demonstrates efficacy in hindering the growth of orthotopic xenograft tumors and increasing survival rates in in vivo models. Overall, ASncmtRNA knockdown is a prospective and clinically relevant treatment strategy for ccRCC. To provide a more comprehensive understanding, the following analysis could be incorporated:

1. Any potential synergistic effects when combining ASncmtRNA knockdown with existing treatments.

2. What are the underlying molecular mechanisms and pathways affected by ASncmtRNA knockdown in primary and metastatic ccRCC.

3. Any potential off-target effects of ASO-1537, any limitations and potential challenges of ASncmtRNA knockdown.

4. The manuscript briefly mentions the reduction in metastatic potential, it needs to elaborate on the specific functional assays or markers used to evaluate metastasis.

Minors:

1. Rearrange the annotations in Figure 2a and 3a.

2. Correct the ASO-1537 dosage in Figure 4a and 5a.

3. Clarify that ASO-1537 dosage was administered as 100 µg per mouse and was not adjusted based on body weight.

Author Response

The manuscript investigates the therapeutic potential of mitochondrial non-coding RNAs (ASncmtRNAs) knockdown in clear cell renal carcinoma (ccRCC). The study focuses on primary cultures derived from ccRCC patient tumors, providing a more clinically relevant model than traditional cell lines. Both primary and metastatic ccRCC, exhibit a pronounced blockage in proliferation and undergo apoptotic death upon ASncmtRNA knockdown, which were consistent with previous findings in tumor cell lines. Furthermore, ASncmtRNA knockdown demonstrates efficacy in hindering the growth of orthotopic xenograft tumors and increasing survival rates in in vivo models. Overall, ASncmtRNA knockdown is a prospective and clinically relevant treatment strategy for ccRCC.

Author response: We thank the reviewer for his or her valuable insight concerning our work. We have addressed the concerns raised by the reviewer as follows:

To provide a more comprehensive understanding, the following analysis could be incorporated:

  1. Any potential synergistic effects when combining ASncmtRNA knockdown with existing treatments.

Author response: The concern raised by the reviewer is quite insightful. On this note, we did indeed perform experiments comparing ASO-1537 with the first-line ccRCC treatment drug, Sunitinib. As shown in the new Supplementary Figure S2, ASncmtRNA knockdown displays a higher efficiency after 48 h of treatment in PT2 and MT1 primary cultures. Moreover, we have also added a new Supplementary Figure S8, which shows that ASO-1537 and Sunitinib, administered individually, do not display a significant difference in survival. A combination of ASO-1537 and Sunitinib, however, elicits a slight improvement in survival compared to each treatment on its own. We have added an analysis of this point in the Discussion section.

  1. What are the underlying molecular mechanisms and pathways affected by ASncmtRNA knockdown in primary and metastatic ccRCC.

Author response: We thank the reviewer for this very important question. Our present study was focused on in vitro and pre-clinical application of ASncmtRNA knockdown for treatment of ccRCC. However, the reviewer may refer to our previous publications, especially the most recent one (Bendek et al., 2023, doi: 10.3390/ncrna9050059), in which we established that the knockdown treatment affects breast tumor and normal cells in an inverse manner concerning genes that are involved in the maintenance of genomic integrity, namely Aurora Kinase A and Topoisomerase IIa. These two genes are upregulated in normal breast epithelial cells upon the treatment, and downregulated in two breast adenocarcinoma cell lines, MDA-MB-231 (triple-negative, highly metastatic) and MCF7 (luminal, tumorigenic but non-invasive). As we hypothesize in that article, this inverse behavior should offer genomic protection for normal cells and should, conversely, stimulate genomic instability in tumor cells in a manner which induces massive apoptotic cell death. Another very important protein that has been found to be downregulated by the treatment in several of our publications on different cell models is survivin (BIRC5), a member of the inhibitor of apoptosis protein (IAP) family, whose downfall should prevent cell cycle progression while, at the same time, should stimulate apoptosis (see ref 20 in the manuscript). These molecular changes in tumor cells from various tissue origins, including key cell cycle progression factors (please see references in the manuscript), should be at the center of the selective effects on tumor cells elicited by the knockdown of ASncmtRNAs. We have hypothesized that these effects could be underlain by an increase in mitochondrial microRNAs (miRNAs) contained in ASncmtRNA-2, induced by the knockdown treatment (Vidaurre et al., 2014 doi: 10.1074/jbc.M114.558841; Bianchessi et al., 2015 doi: 10.1016/j.yjmcc.2015.01.012; Fitzpatrick et al., 2019 doi: 10.1038/s41419-019-1649-3; Farfán et al., 2021 doi: 10.1186/s40659-021-00356-0). These miRNAs could putatively target the key genes that are affected by the treatment, thus causing the effects on proliferation and cell death that we have observed in several tumor cell models (see references in the manuscript). We have now included this information in the Discussion section.

  1. Any potential off-target effects of ASO-1537, any limitations and potential challenges of ASncmtRNA knockdown.

Author response: This is a very pertinent concern raised by the reviewer. As stated in the manuscript, we previously observed no detrimental effects of the treatment in a pre-clinical mouse model of melanoma (Lobos-González et al., 2016, doi: 10.18632/oncotarget.11110). In this work we showed that the treatment did not elicit inflammatory response and did not affect body weight. In addition, we performed a Phase Ia Clinical Trial at UCSF, which was well-tolerated up to the highest dose (800 mg per patient) and showed no toxicity (Clinical Trial NCT02508441; Dhawan et al.,2018, J. Clin. Oncol. 36:2557). A plausible limitation would concern the uptake of ASO-1537 into different organs and tissues. In this respect, we have shown that a fluorescently-labeled ASO against mouse ASncmtRNA is efficiently incorporated into renal tumor tissue (Borgna et al., 2017 doi: 10.18632/oncotarget.18460). To date, we have not performed in vivo assays targeting specific tissues, but this aim could be achieved through tissue-specific nanoparticles of different chemical nature. However, our present work represents a proof-of-concept study of the efficacy of ASncmtRNA knockdown for the reduction of ccRCC tumorigenic capacity. We have incorporated this analysis into the Discussion section.

  1. The manuscript briefly mentions the reduction in metastatic potential, it needs to elaborate on the specific functional assays or markers used to evaluate metastasis.

Author response: As stated in the manuscript, the decrease in metastatic potential was reflected by the lower levels of N-cadherin in ASO-1537-treated mice. N-cadherin is an epithelial marker and its reduction is a sign of loss of anchorage to the epithelial tissue or origin and consequential migration through the surrounding mesenchymal tissue, during the process of epithelial-mesenchymal transition (EMT), which is the first step in metastasis. Although our manuscript focuses on tumorigenic potential (as per the title and abstract) rather than the metastatic process, we believe that the reduction in N-cadherin levels reflects a decrease in invasiveness of the ccRCC cells, as reported in previous publications for other tissue types (Lobos-González et al., 2016 doi: 10.18632/oncotarget.11110; Borgna et al., 2017 doi: 10.18632/oncotarget.18460; Borgna et al., 2020 doi: 10.7150/jca.38880). Finally, as mentioned in the text, N-cadherin is also involved in proliferation, which is the reason for including this marker in Fig. 3.

Minors:

  1. Rearrange the annotations in Figure 2a and 3a.

Author response: We thank the reviewer for this observation. Some of the annotations were misaligned due to the conversion to pdf. We have now rearranged them in order to improve the display quality.

  1. Correct the ASO-1537 dosage in Figure 4a and 5a.

Author response: We are not sure what the reviewer is referring to by “correcting the dosage”.

  1. Clarify that ASO-1537 dosage was administered as 100 µg per mouse and was not adjusted based on body weight.

Author response: As stated in the text and the figures, the dose was 100 mg per mouse and not adjusted to mouse weight. The mice were 20-25 g in weight and all mice received the same dosage.

Round 2

Reviewer 2 Report

Comments and Suggestions for Authors

The authors have sufficiently improved the manuscript. I endorse its acceptance.